

# Spatial layout optimization model integrating layered attention mechanism in the development of smart tourism management

Jie Ding[1], Lingyan Weng[1], Lili Fan[1] and Peixue Liu[2]

[1] School of Tourism, Nanjing Institute of Tourism and Hospitality, Nanjing, Jiangsu, China
[2] School of Business Administration, Nanjing University of Finance and Economics, Nanjing, Jiangsu, China

## ABSTRACT

Tourism demand projection is paramount for both corporate operations and destination management, facilitating tourists in crafting bespoke, multifaceted itineraries and enriching their vacation experiences. This study proposes a multi-layer self attention mechanism recommendation algorithm based on dynamic spatial perception, with the aim of refining the analysis of tourists' emotional inclinations and providing precise estimates of tourism demand. Initially, the model is constructed upon a foundation of multi-layer attention modules, enabling the semantic discovery of proximate entities to the focal scenic locale and employing attention layers to consolidate akin positions, epitomizing them through contiguous vectors. Subsequently, leveraging tourist preferences, the model forecasts the likelihood of analogous attractions as a cornerstone for the recommendation system. Furthermore, an attention mechanism is employed to refine the spatial layout, utilizing the forecasted passenger flow grid to infer tourism demand across multiple scenic locales in forthcoming periods. Ultimately, through scrutiny of data pertaining to renowned tourist destinations in Beijing, the model exhibits an average MAPE of 8.11%, markedly surpassing benchmarks set by alternative deep learning models, thereby underscoring its precision and efficacy. The spatial layout optimization methodology predicated on a multi-layer attention mechanism propounded herein confers substantive benefits to tourism demand prognostication and recommendation systems, promising to elevate the operational standards and customer contentment within the tourism sector.

## INTRODUCTION

Recent years have witnessed a rapid rise in China's tourist industry, propelled by the country's economic growth and the rapid expansion of the Internet (*Liu, Gao & Li, 2024*). Moreover, China boasts a wealth of tourism resources that support the industry's development and contribute to the steady increase in the number of tourists visiting the country. With the increasing number of tourists, this has also brought about some

Corresponding author
Peixue Liu, 9120221002@nufe.edu.cn

challenges to the development of the tourism industry. The non-persistence and perishability of certain tourism products have resulted in the waste of tourism resources, such as unsold air tickets and hotel rooms that cannot be stored. Additionally, there has been an imbalance in the number of tourists and resource allocation at tourist attractions. To address these challenges, accurate prediction of tourism demand is essential. This will enable tourism professionals to allocate resources more effectively and help government agencies and tourism companies formulate relevant policies and invest in infrastructure. Therefore, accurate prediction of tourism demand plays a crucial role in the development of the tourism industry (*Fan, Yin & Zha, 2024*).

Tourism volume forecast is a complicated nonlinear issue that is readily impacted by a wide range of variables, including seasonality, natural changes, holidays, and crises. The demand for tourism is erratic and unpredictable due to these variables, which makes forecasting extremely challenging. The conventional approach to forecasting tourism demand mostly uses data from monitoring equipment or historical information made public by government research agencies, although both approaches have limitations. The government's historical data is frequently delayed, has inadequate sample sizes, and does not accurately depict the dynamic features of passenger flow (*Bufalo & Orlando, 2024*). On the other hand, obtaining data through the use of monitoring equipment is expensive, complicated to execute, and prone to large inaccuracies. For these reasons, developing a reliable model to forecast tourism demand is a difficult undertaking (*Li et al., 2024*).

There are various models available for predicting nonlinear, non-stationary, and complex tourism demand, including traditional time series models, econometric models, artificial intelligence models, and hybrid models (*Zheng, Li & Li, 2024*). When historical data is consistent and adequate, time series models excel at capturing patterns and periodic changes in the data. The most frequently employed time series models are AR, ARMA, and ARIMA. On the other hand, econometric models provide more detailed economic analysis and forecasts by considering the influence of macroeconomic factors on tourism demand. Artificial intelligence models, while adept at handling complicated data patterns and nonlinear interactions, are sometimes regarded as 'black boxes' due to their internal workings requiring further justification and verification.

To the best of the author's knowledge, not many research have used geographical data to estimate traveler demand. Spatial lag factors have been included into time series or econometric models in *Cook, Hays & Franzese (2023)* to create spatiotemporal autoregressive models. This model incorporates the temporal and spatial correlation of demand for tourism, which can more precisely capture the influence of location on demand (*Cook, Hays & Franzese, 2023*; *Yang & Zhang, 2019*). Because it takes into account the interaction and effect of nearby geographic locations, the creation of a spatiotemporal autoregressive model can improve the prediction results' accuracy and dependability. When it comes to tourist demand forecasting, this model excels at capturing demand patterns across several locations, which empowers decision-makers to create more effective tourism development strategies and resource allocation plans. However, although traditional models have achieved good prediction results, their non-linear processing ability is limited, making it difficult to avoid error accumulation. Moreover, these

prediction methods are only reflected in time. In tourism demand forecasting, spatial correlation not only exists between scenic spots, but also between scenic spots with similar cultural backgrounds but distant distances, as well as between scenic spots and surrounding areas (such as hotels and restaurants). The existence of this spatial correlation is crucial for accurately predicting tourism demand, as it reflects the impact of different geographical locations on tourism demand and the correlation between scenic spots and surrounding resources.

To overcome the above problems, this article proposes a multi-layer self attention mechanism recommendation algorithm based on dynamic spatial perception. The main contributions of this article are as follows:

By introducing a multi-layer self attention mechanism, our model can more comprehensively analyze and understand the complex relationship between users and tourist destinations. This multi-layer self attention mechanism allows us to pay attention to important information at different levels in the data layer by layer, effectively capturing more detailed correlations and interaction patterns between users and tourist destinations.

The introduction of dynamic perception algorithms combines the concept of dynamic spatial perception in our tourism prediction model, which is of great significance for improving the accuracy and practicality of tourism prediction. This algorithm can flexibly adjust attention weights based on the changes of users and tourist destinations in different time and space, in order to better adapt to the actual tourism situation.

## RELATED WORK

With the increasing recognition of the importance of tourism industry prediction, many studies have focused on improving prediction techniques to enhance prediction effectiveness. The methods used for predicting tourism demand can generally be divided into two categories: qualitative research and quantitative research. Qualitative research methods such as the Delphi method focus on expert judgment and subjective evaluation, which have certain requirements for accurate judgment of the tourism market, but have poor generalization ability (*Wu, Song & Shen, 2017*). On the contrary, quantitative research focuses more on using data and mathematical models for prediction, with stronger generalization ability and objectivity. In tourism industry forecasting, quantitative research methods are usually more favored because they can use historical data and statistical models to analyze trends, patterns, and correlations, thus making more accurate and reliable predictions. Therefore, most current research on tourism demand prediction mainly focuses on quantitative methods, which rely on historical tourist arrival data and various tourism demand prediction factors to construct models (*Shareef et al., 2024*).

Two research stages are often distinguished in the development of tourist demand forecast methods: the classical time series model and econometric tourism demand prediction research.

Time series prediction models have a long history and have made significant progress in forecasting tourism demand (*Yang et al., 2022*). As a result, among the various forecast models for tourism demand, the time series prediction model is now considered reasonably reliable. With just one dataset needed, a time series model can forecast future

patterns by analyzing historical trends from previous time series data. These models are often simple yet effective, as they aim to identify patterns, trends, and durations within time series data (*Bharadiya, 2023*). The Seasonal Autoregressive Integrated Moving Average (SARIMA) approach was employed by the author in *Arshad et al. (2023)* to forecast the travel and tourism sector in India from March 2020 to December 2020. Next, assess the suggested model's robustness by contrasting its outcomes with those of the Holt Winter (H–W) model. *Neves, Nunes & Fernandes (2022)* created a seasonal autoregressive comprehensive moving average time series model for tourism demand using Saar Island, Cape Verde, as an example. This technique yielded accurate predictions for 2019 when applied to the time series of monthly lodging days in popular tourist spots on Saar Island, Cape Verde, from January 2000 to December 2018.

Unlike time series models, econometric models are primarily used to capture causal relationships between tourist numbers and various explanatory factors. These explanatory factors can include but are not limited to income level, tourism prices, tourism convenience, destination image, transportation conditions, *etc*., (*Gunter & Zekan, 2021*). These factors may affect tourists' travel motivation and behavioral choices. Time-varying parameter models (TVPM), autoregressive distributed lag models (ARDL), and vector autoregressive models (VARM) are a few common econometric models. The VAR model can simultaneously consider the interrelationships between multiple variables and is suitable for analyzing the linkage effects of multiple explanatory factors on tourist numbers (*Adedoyin & Bekun, 2020*), the ARDL model is suitable for analyzing the causal relationship between long-term and short-term variables, and can be used to explore the long-term and short-term effects of explanatory factors on tourist numbers (*Altaf, Awan & Rehman, 2023*; *Wan & Song, 2018*), and TVPM can consider the variation of factor parameters over time, which has certain advantages in exploring the impact of explanatory factors on tourist numbers over time (*Liu et al., 2024*). Compared with time series models, econometric models focus more on explaining causal relationships and the mutual influence between variables, rather than just fitting and predicting time series data. Therefore, selecting the appropriate model needs to be determined based on specific research questions and data characteristics. Meanwhile, research comparing time series models and econometric models suggests that prediction results may vary depending on the environment and data (*Peng, Song & Crouch, 2014*).

Although time series models and econometric models have initially achieved some results in the field of tourism demand forecasting, with the continuous research of scholars on tourism demand forecasting, the accuracy of demand forecasting continues to improve. Therefore, these two types of model prediction methods are far from meeting the current research on tourism demand prediction. In addition, the current tourism data is complex and irregular, and to a certain extent, the data has a high degree of nonlinearity, which also makes time series prediction models unable to handle the current tourism demand prediction task (*Sun et al., 2023*). Attention mechanism is an important technique in deep learning, which can help models focus on important parts and ignore irrelevant information when processing sequence data. This mechanism mimics the selective

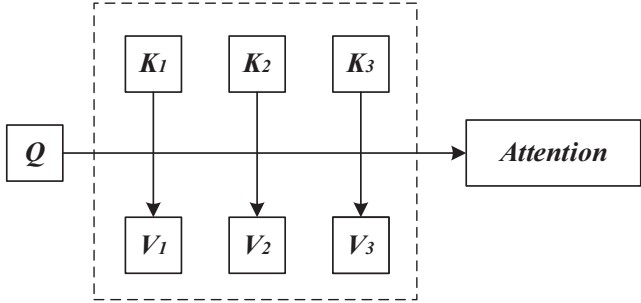

**Figure 1** **Attention mechanism framework.**

attention process of humans in processing information, thereby improving the performance of deep learning models (*Kim et al., 2021*; *Adil et al., 2021*; *Liu et al., 2022*).

## METHODOLOGY

### Attention mechanism

In the field of tourism recommendation, attention mechanisms have also received widespread attention and successful applications. Similar to applications in natural language processing and machine translation, attention mechanisms in tourism recommendations mimic human visual principles, selectively focusing on various factors related to tourism and extracting important information, while ignoring factors that have a relatively small impact on the final recommendation results. By learning the importance of various elements in the data for the final tourism recommendation, the attention mechanism assigns different weights to each element. This means that in tourism recommendation, factors with higher weights will have a greater impact on the final tourism recommendation, while factors with lower weights will have a smaller impact on the final recommendation. The attention mechanism model is shown in Fig. 1.

In Fig. 1, Q represents Query, K represents Key, and V represents Value. The attention mechanism describes the problem as a given target element query, calculates the similarity between the query and the key, and finally multiplies the value to obtain the final result. The formula for calculating attention a is as follows:

$$a = \sum_{i=1}^{l} f(Q, K)V \tag{1}$$

where l represents the number of samples, f represents the function for calculating query and key similarity. Common methods include dot product, cosine similarity, and multi-layer neural networks.

Self attention network is a further improved model of attention mechanism, which determines the user's focus on information by calculating the interdependence between different elements in a sequence, and calculates the user's interest preference score based on this. The core operation of self attention mechanism is the scaling dot product operation, as shown in Fig. 2.

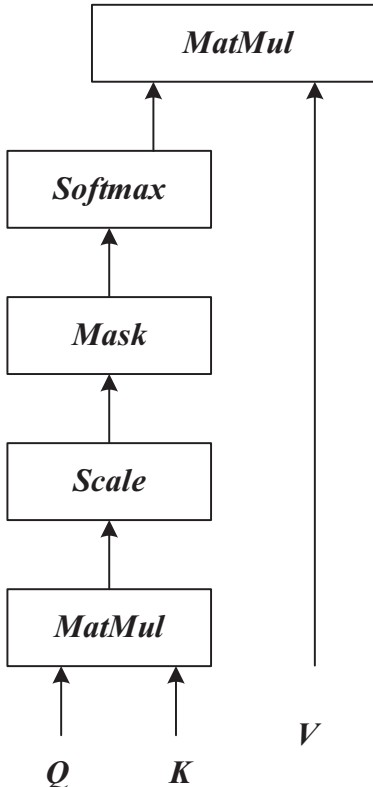

**Figure 2 Scaling dot product operation of self attention mechanism.**

The self attention network embeds queries (Q), keys (K), and values (V) into the same dimensional space, then performs dot multiplication on the queries and keys, and performs scaling operations to obtain the similarity between the queries and keys through the softmax function. Finally, the similarity is weighted and multiplied with the value to obtain the attention score. The specific formula is as follows:

$$Attention(Q, K, V) = softmax\left(\frac{QK^T}{\sqrt{d_k}}\right) V \tag{2}$$

where $d_k$ is the dimension of the query and key.

In addition to scaling dot product operations, self attention networks also apply a series of techniques to ensure smooth training, including:

Feed Forward Network (FFN): Although the self attention mechanism can use adaptive weights to aggregate embeddings of all previous attractions, it ultimately remains a linear model. In order to endow the model with nonlinear features and consider the interactions between different potential attractions, the self attention mechanism will process the obtained attention through a double-layer feedforward network. It is worth noting that these multi-layer feedforward networks share parameters in the tourism recommendation model. The specific formula is as follows:

$$Output = ReLU(W_2 \cdot ReLU(W_1 \cdot Attention(Q, K, V) + b_1) + b_2) \tag{3}$$

where $\text{Attention}(Q, K, V)$ represents the attention output obtained by the self attention mechanism, $W_1$ and $W_2$ are the weight parameters of the double-layer feedforward network, $b_1$ and $b_2$ are bias parameters, respectively, and ReLU is the modified linear unit activation function. In this way, by introducing a dual-layer feedforward neural network, the tourism recommendation model has gained more complex nonlinear modeling capabilities to better capture the intricate relationships between users and different tourist attractions.

Residual connections (RC): In some cases, multi-layer neural networks have been proven effective in learning travel recommendation features. However, simply adding more layers does not always lead to better performance. The core idea of residual connection is to solve this problem by transmitting low-level features to higher levels. Therefore, if the underlying features are useful, the model can more easily pass them on to the final layer. In the context of tourism recommendations, we also assume that residual connections are useful. The specific formula is as follows:

$$RC(o) = 0 + e \tag{4}$$

where Among them, o is the output of the network before residual connection, and e is the embedding vector of the last sample in the input sequence.

Layer normalization (LN): Adjusting it to zero mean and unit variance can help improve the stability of the neural network and accelerate the training process. In layer normalization, the statistical data used is based on the features in the current batch rather than the entire dataset. Specifically, for the input vector x, which contains an example of all features, the layer normalization operation is defined as:

$$LayerNorm(x) = \alpha \cdot \frac{x - \mu}{\sqrt{\sigma^2 + \varepsilon}} + \beta \tag{5}$$

where $\cdot$ represents dot product operation, $\mu$ is the mean of input vector x, $\sigma$ is the variance of input vector x, $\alpha$ and $\beta$ are scaling factors and bias terms.

Dropout: During the training process, neurons are randomly closed with a certain probability, and all neurons are used during testing to alleviate overfitting problems and improve the model's generalization rate.

## Multilayer self attention mechanism recommendation algorithm based on dynamic spatial perception (MSAMR-DSP)

The dynamic characteristics of tourist attractions are crucial in recommendation tasks. Researchers typically focus on how to capture users' changing interests over time from historical interaction sequences, while ignoring the constantly evolving hidden features of attractions. Firstly, the user's historical interaction sequence not only includes the dynamic changes in user interests over time, but also contains information on the evolution of potential features of scenic spots over time. If the potential features of tourist attractions are considered invariant, this information will be lost when constructing the association between users and items, resulting in incomplete predictions. Therefore, it is not appropriate to only explore the dynamic changes in the sequence from the user's

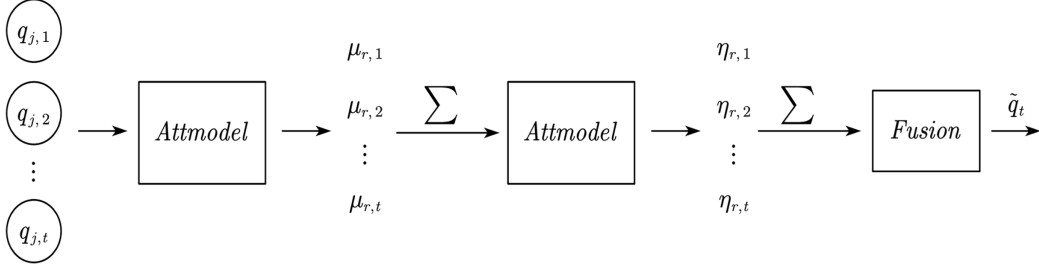

**Figure 3 Project level attention model.**

perspective. Secondly, most sequence models only focus on the implicit feedback sequence of individual users, while ignoring the collaborative relationships between other users. Analyzing the dynamic changes in the sequence from the perspective of scenic spots can enhance the collaborative effect between users in the model to a certain extent, thereby more accurately analyzing users' intentions.

Therefore, the method proposed in this article uses a self attention network module to obtain patterns of item feature transformation between different users. Then, the obtained dynamic feature representation of the scenic spot and the traditional user dynamic feature representation are input into the neural network to obtain its complex interaction pattern, achieving a more complete prediction. Therefore, we propose a multilayer self attention mechanism recommendation algorithm based on dynamic spatial perception (MSAMR-DSP), which overcomes the problem of previous recommendation systems ignoring the dynamic characteristics of items. It models personalized sequence patterns between users and items from two levels: user dynamic interest features and item dynamic features; Secondly, it solves the problem of model dynamic sensitivity caused by modeling both item and user dynamic features simultaneously. By balancing the dynamic and static features of the model, it makes the training more stable.

The application level of attention mechanism in current recommendation systems is relatively single, and the mining depth of multiple factors is shallow, failing to fully consider the impact of multi factor attention mechanism and the fusion weight of various indicators on the system. On the basis of the above research, we introduced multi-dimensional and multi-level attention modeling, mining more comprehensive and accurate weights of various types of information in the model. The following text will introduce attention mechanism models and their applications at different levels. The network structure is shown in Fig. 3.

This study models attention mechanisms through multi-layer perceptrons, and incorporates the representation vector q of user u's historical interaction items q into the attribute preferences of the recommended item r, which can be obtained through Eq. (1):

$$q_j = \sum_{t=1}^{t} \mu_{j,t} q_{j,t} \tag{6}$$

where $\mu_{j,t}$ is the preference of the recommended project $q_r$ for the t-th attribute $q_j$ of the historical interaction project, which is defined as:

$$\mu_{j,t} = \frac{exp(score(q_{r,t}, q_{j,t}))}{\left[\displaystyle\sum_{t=1}^{n\sum_{r,t} j,t^{\lambda}} exp\right]} \tag{7}$$

In recommendation systems, there is a significant difference in the historical interaction data for different items. The traditional softmax function may excessively penalize users with longer historical interaction information. Therefore, this article introduces a smoothing exponent $\lambda$ with $0 \leq \lambda \leq 1$. The attention on overly active parts of the system will not be excessively penalized. In this case, q is the alignment score learned by the model. In this study, an additive (Concat) method was used to calculate the alignment score $score(q_{r,t}, q_{j,t})$, and a hyperbolic tangent function (tanh) was used as the nonlinear activation function. The calculation method for $score(q_{r,t}, q_{j,t})$ is as follows:

$$score(q_{r,t}, q_{j,t}) = \mathbf{v}_{a1}^T \tan h(\mathbf{W}_{a1}[q_{r,t}; q_{j,t}]) \tag{8}$$

where $\mathbf{v}_{a1}$ and $\mathbf{W}_{a1}$ are learnable parameters that participate in training as part of the overall model. By Eq. (6), we obtain the representation vector $\mathbf{q_j}$ of user u's historical interaction item $q_j$ integrated with the attribute preference of the recommended item $q_r$. Furthermore, by introducing an additional layer of attention mechanism, we capture the interaction vector $\bar{\mathbf{q}}_\mathbf{j}$ of the recommended item with respect to the attention weight $\alpha_{r,j}$ obtained from the model above, where $q_j \in I_u$ and $I_u$ represents the set of historical interaction items of user u. The structure of the model is similar to the aforementioned process, and the calculation of the new layer of attention weight $\alpha_{r,j}$ is as shown in equation:

$$\eta_{r,j} = \frac{exp(score(\bar{q}_r, q_j))}{\left[\displaystyle\sum_{t=1}^{n\sum_{r} j^{\lambda}} exp\right]} \tag{9}$$

The alignment score a still uses a multi-layer perceptron model for learning, using tanh as the nonlinear activation function, which is defined as shown in equation:

$$score(\bar{q}_r, q_j) = \mathbf{v}_{a2}^T tan h(\mathbf{W}_{a2}[\bar{q}_r; q_j]) \tag{10}$$

Similarly, $\mathbf{v}_{a2}$ and $\mathbf{W}_{a2}$ are learnable parameters that can be obtained through model training.

Next, we constructed an attention model using user and social information, aiming to explore the attention relationship between user u and their trusted user v. The network structure is similar to that shown in Fig. 4.

$\mathbf{p}_u$ is the embedding vector of user u, and the trusted user set of user u is $s_u = \{s_1, s_2, \ldots, s_m\}$. In this model, a multi-layer perceptron network is introduced to learn the attention relationship between users and their trusted friends. Although most MLPs use the sigmoid function as the activation function, due to the high risk of gradient vanishing

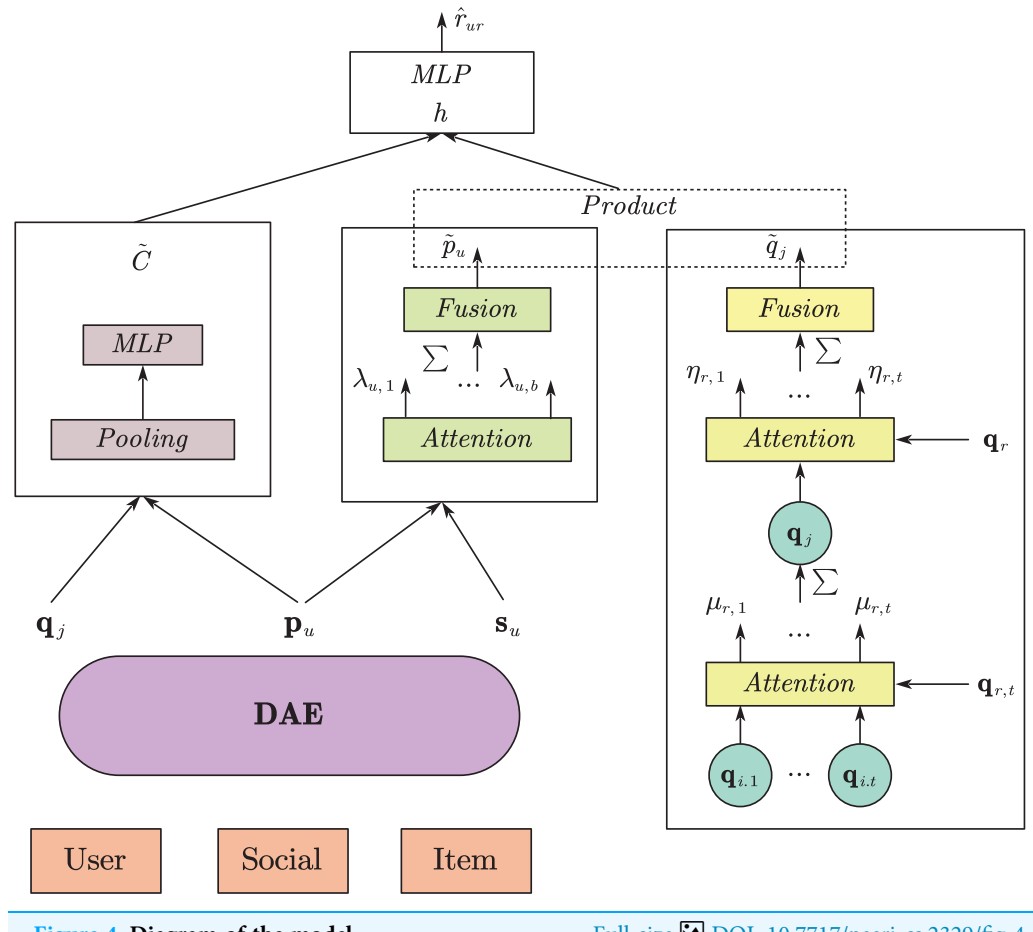

**Figure 4** **Diagram of the model.** 

and the large amount of derivative calculation during backpropagation, ReLU is used as the nonlinear activation function of the model in this study. The alignment score $score(p_u, s_m)$ of user u for its trusted user m is obtained by the following formula:

$$score(p_u, s_m) = \text{RELU}(\dots\text{RELU}\mathbf{W}_2(\text{RELU}(\mathbf{W}_1[p_u, p_m] + b_1) + b_2)\dots) \tag{11}$$

where $\mathbf{W} = [\mathbf{W}_1, \mathbf{W}_2]$ with $\mathbf{W}_1 \in R^{d \times h}$ and $\mathbf{W}_2 \in R^{d \times 1}$ (d is the dimension of input features, and h is the number of neurons in the hidden layer) is the weight matrix of the attention model. and $b_1 \in R^h$ and $b_2 \in R$ are the bias vectors of the system. Next, normalize the alignment score a as follows:

$$\lambda_{u,m} = \frac{\exp(score(\mathbf{p_u}, \mathbf{s_m}))}{\left[\sum_{m \in s_u} \exp(score(\mathbf{p_u}, \mathbf{s_m}))\right]^{\lambda}} \tag{12}$$

Similar to the above model, in order to fully integrate the user's preference for trusting users while considering the user's original information, the definition of the user's comprehensive feature representation $\tilde{\mathbf{P}}_\mathbf{u}$ is as follows:

$$\tilde{\mathbf{P}}_{\mathbf{u}} = \alpha\mathbf{p}_{\mathbf{u}} + (1 - \alpha) \sum_{m \in \mathbf{s}_u} \lambda_{u,m}\mathbf{s}_{\mathbf{m}} \tag{13}$$

where $\alpha$ is a balance parameter used to balance between the user's original features and the trusted user embedding. The parameter $\alpha$ controls the proportion of the weighted sum of the original feature $\mathbf{p}_{\mathbf{u}}$ of user u and its trusted user embedding $\mathbf{s}_{\mathbf{m}}$ in the comprehensive feature representation. Specifically, when $\alpha$ is large, the comprehensive feature representation of user u relies more on its own original features; When $\alpha$ is small, the comprehensive feature representation relies more on the embedding of its trusted users. By adjusting the value of $\alpha$, the model can find an optimal balance between utilizing the user's own characteristics and trustworthy user characteristics, thereby enhancing the expressive power of the user's comprehensive feature representation.

In recommendation systems, the inner product of user and item implicit feature vectors is usually used for rating prediction. In this study, after using a multi-layer attention model, the user comprehensive feature representation $\tilde{\mathbf{P}}_{\mathbf{u}}$ and the item comprehensive feature representation $\tilde{\mathbf{P}}_{\mathbf{r}}$ were obtained, respectively. Therefore, the specific form of rating prediction for the recommended item r by user u is as follows:

$$\tilde{r}_{ur} = \tilde{\mathbf{P}}_{\mathbf{u}} \cdot \tilde{\mathbf{P}}_{\mathbf{r}} \tag{14}$$

## Model fusion

This study has fully considered the multi-layer attention information in recommendation problems, including: using a double-layer attention network to simultaneously mine the preferences of the recommended items for historical items and their item attributes, and mining the attention weights of the target users towards their trusted users under a new social trust relationship, which brings rich auxiliary information to this study and alleviates the problem of data sparsity in the system. Excessive mining of basic information may result in the loss of some key features. Therefore, based on the above attention model, this study uses a multi-layer perceptron to mine the original implicit features of users and items. A simple pooling layer is used to concatenate the user embedding vector and item embedding vector, as shown in Eq. (15):

$$Con(\mathbf{p}_{\mathbf{u}}, \mathbf{q}_{\mathbf{r}}) = \begin{bmatrix} \mathbf{p}_{\mathbf{u}} \bullet \overline{\mathbf{q}_{\mathbf{r}}} \\ \mathbf{p}_{\mathbf{u}} \\ \mathbf{q}_{\mathbf{r}} \end{bmatrix}. \tag{15}$$

Similar to previous studies, multilayer perceptrons still use the ReLU function as the activation function, as shown in Formula (16):

$$\tilde{C}(\mathbf{p}_{\mathbf{u}}, \mathbf{q}_{\mathbf{r}}) = \text{RELU}(\dots\text{RELU}\mathbf{W}_2(\text{RELU}(\mathbf{W}_1 Con(\mathbf{p}_{\mathbf{u}}, \mathbf{q}_{\mathbf{r}}) + b_1) + b_2)\dots) \tag{16}$$

After adjusting the vector dimension, the obtained implicit features of the original information and the score prediction of the fused attention obtained in "Multilayer Self Attention Mechanism Recommendation Algorithm Based on Dynamic Spatial Perception

(MSAMR-DSP)" are element fused to obtain a comprehensive variable, as shown in Eq. (17):

$$h = \tilde{r}_{ur} \odot \tilde{C}_{ur}. \tag{17}$$

At the top of the model, there is also a multi-layer perceptron used for the final rating prediction of the system. To prevent overfitting, the Dropout mechanism is introduced, which reduces the homogenization problem in the multi-layer perceptron by causing some neurons to lose contact with probability $p$. After processing by the Dropout mechanism, the comprehensive variable is obtained. During the training process, the backpropagation algorithm BPTT is used for model optimization, and its cross entropy loss function is as follows:

$$Loss = -\sum_{u=1}^{m}\sum_{i=1}^{n} r_{ur} \log \tilde{h} + (1 - r_{ur}) \log(1 - \tilde{h}). \tag{18}$$

This article has constructed attention models for project attributes and project interaction levels, as well as user social attention models and basic information mining models. The various parts have been integrated to obtain rating predictions that incorporate richer information. This model adjusts weights by introducing temporal and spatial feature embeddings to adapt to dynamic changes in users and travel destinations. Specifically, temporal feature embedding includes seasonal changes, time period changes, and holiday effects, while spatial feature embedding covers geographic location and real-time weather conditions. The model utilizes an embedding layer to transform these features into high-dimensional vectors and fuse them with other features to form a joint feature vector for dynamically adjusting weights. The overall framework of this study is shown in Fig. 4.

In this article, we choose ReLU as the activation function in the MLP instead of the traditional activation functions sigmoid or tanh, mainly for the following reasons: First, ReLU's computation is very simple, requiring only a threshold operation, which makes its computational efficiency high during both forward and backward propagation. This helps speed up model training and prediction. Sigmoid and tanh activation functions require the computation of exponential functions, which is computationally expensive, especially in deep networks, significantly increasing training time. Second, ReLU can effectively handle nonlinear relationships because it is linear in the positive region and zero in the negative region, allowing it to introduce nonlinear transformations and thus enhance the model's expressive power. While sigmoid and tanh can also introduce nonlinearity, their gradient saturation properties limit their effectiveness in deep networks. Finally, for deeper networks, the choice of ReLU activation function is particularly critical because it can better alleviate the vanishing gradient problem, thus supporting deeper neural network structures. In shallow networks, sigmoid and tanh may still be usable, but in deep networks, the advantages of ReLU are more pronounced.

| Table 1 Dataset is divided in detail. | | | |
|---|---|---|---|
| Dataset | Period | Training set | Test set |
| Beijing | June 1 to June 29, 2016 | 1,520 | 153 |

# EXPERIMENTAL RESULT

## Experimental preparation

This study uses mobile signal data from the popular tourist destination of Beijing as a sample for an empirical investigation of the spatiotemporal grid of tourism demand, which aims to validate the model's efficacy. The spatiotemporal location data of all users, including locals and visitors as well as other groups, is included in the mobile signaling data that is provided by China Mobile (https://www.statista.com/statistics/272097/customer-base-of-china-mobile/, doi: 10.18653/v1/2022.seretod-1.7) and China Unicom (https://zenodo.org/records/11827345, doi: 10.5281/zenodo.11827345). Online data collection also includes geographic information maps of cities and the locations of key attractions. Furthermore, we used the K-fold cross validation method to keep each sub sample as the test set for model validation and the remaining nine sub samples as the training set for model training in order to fully validate the performance of various models. Each dataset was randomly divided into 10 equal sub samples. Table 1 displays comprehensive dataset details.

In the meanwhile, we collected statistics on passenger volume from six scenic Beijing grids, as shown in Fig. 5. The tourism demand time series for many picturesque locations show nonlinear and cyclical patterns.

The experimental hardware platform is Intel (R) Core (TM) i7-6700 CPU@3.40 GHz (8-core CPU), 16 GB of memory, Intel (R) HD Graphics 530. The experimental software platform is Win10 operating system, and the development environment is Python 3.7 programming language. The Tensorflow library and Scikit learn library in Python are used to build suggested sentiment analysis methods and comparative experiments. According to the experiment, different hyperparameters and settings have been tested. By comparison, the following parameters were ultimately selected: the dimension of the word vector is 300, the learning rate is 0.00001, the dropout value is 0.75, the batch size is 30, the lstm_size is 32, and the iteration count is 1 K.

## Performance evaluations

Mean absolute error (MAE), root mean square error (RMSE), mean absolute percentage error (MAPE), and coefficient of determination (R2) are the four performance measures that are used. Among these, runtime is the amount of time needed for the model to finish executing on the same server without requiring special computations. These are the remaining computations:

$$\text{MAE} = \frac{1}{n}\sum_{i=1}^{n}|x_i - \tilde{x}_i| \tag{19}$$

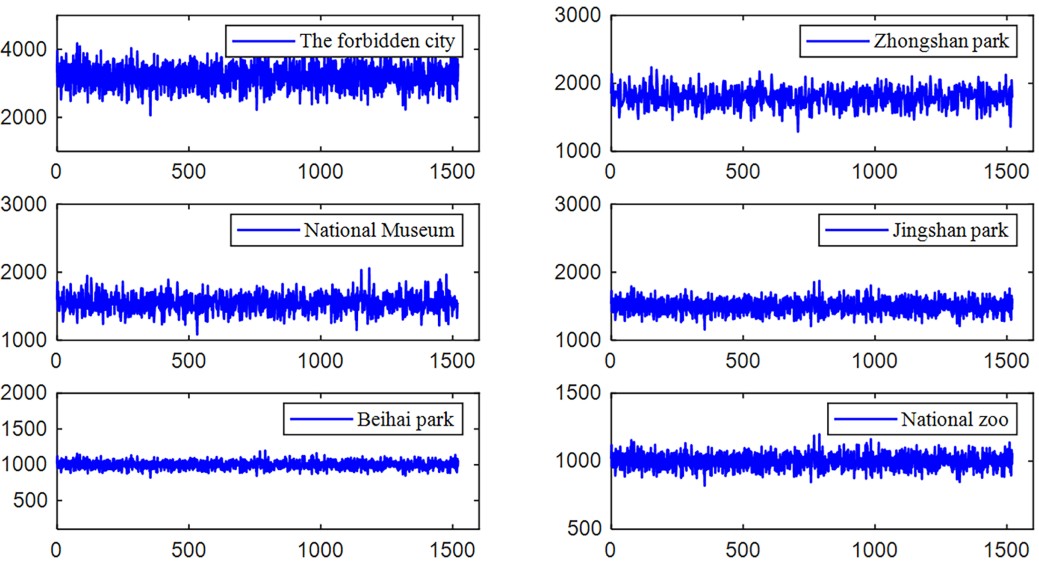

**Figure 5 Selected attractions' tourism demand distribution.**

$$RMSE = \sqrt{\frac{1}{n}\sum_{i=1}^{n}(x_i - \tilde{x}_i)^2} \tag{20}$$

$$MAPE = \frac{1}{n}\sum_{i=1}^{n}\frac{|x_i - \tilde{x}_i|}{x_i} \tag{21}$$

and

$$R^2 = \frac{\sum_{i=1}^{n}(\tilde{x}_i - \bar{x}_i)^2}{\sum_{i=1}^{n}(x_i - \bar{x}_i)^2} \tag{22}$$

where $x_i$, $\tilde{x}_i$, and $\bar{x}_i$ stand for the average of real tourist demand observations, anticipated tourism demand observations, and actual tourism demand observations, respectively. $n$ indicates the number of predicted samples.

In addition, we use two widely used evaluation metrics in the recommendation field, namely Area Under Curve (AUC) and Normalized discounted Cumulative Gain (NDCG). The definition of AUC is as follows:

$$AUC = \frac{\sum_{i \in U^+, j \in U^-} I(P_i, P_j)}{|U^+| \times |U^-|} \tag{23}$$

where $U^+$ and $U^-$ are sets of positive and negative examples in the system test set, respectively. $P_i$ is the predicted probability of project $i$. $I(P_i, P_j) = \begin{cases} 1, & P_i > P_j \\ 0.5, & P_i = P_j \\ 0, & P_i < P_j \end{cases}$.

Generally speaking, $ACU \in [0, 1]$, and the closer its value is to 1, the higher the probability

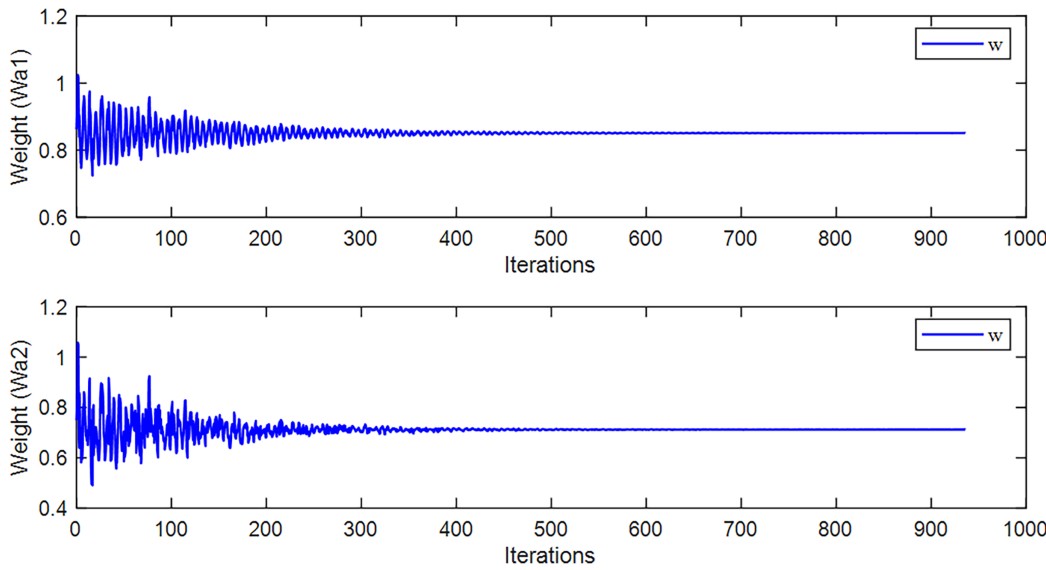

**Figure 6 The weight values during the learning process of the method proposed in this article.**

of positive examples ranking before negative examples, that is, the higher the accuracy of recommendations.

NDCG measures the ranking quality of recommendation lists in TOP-N tasks, defined as:

$$NDCG@N = \frac{DCG}{iDCG}, DCG@N = \sum_{i=1}^{k} \frac{2^{r(l)} - 1}{\log_2(i+1)}. \tag{24}$$

NDCG has a $log_2(i+1)$ discount on the lower ranked items in the recommendation list, where i is the ranking of the items in the list, and iDCG is the ideal DCG value, where the high rated items are exactly at the forefront of the low rated items. The larger the NDCG value, the better the sorting effect of the list obtained by the algorithm in actual testing.

### Experimental result

Firstly, we use the algorithm proposed in this article for training, and during the training process, the weight results are shown in Fig. 6.

Next, we tested the predictive performance and recommendation performance of the algorithm proposed in this article separately. The compared prediction algorithms include TVPM, ARDL, and VARM. The recommended algorithms to be compared are Probabilistic Matrix Factorization (PMF), Neural Collaborative Filtering (NCF), Social-based Rating Prediction (SBRP), and Attention-based Enhanced Adaptive Social Recommendation (AEASR). The experimental results are shown in Figs. 7 and 8, Tables 2 and 3.

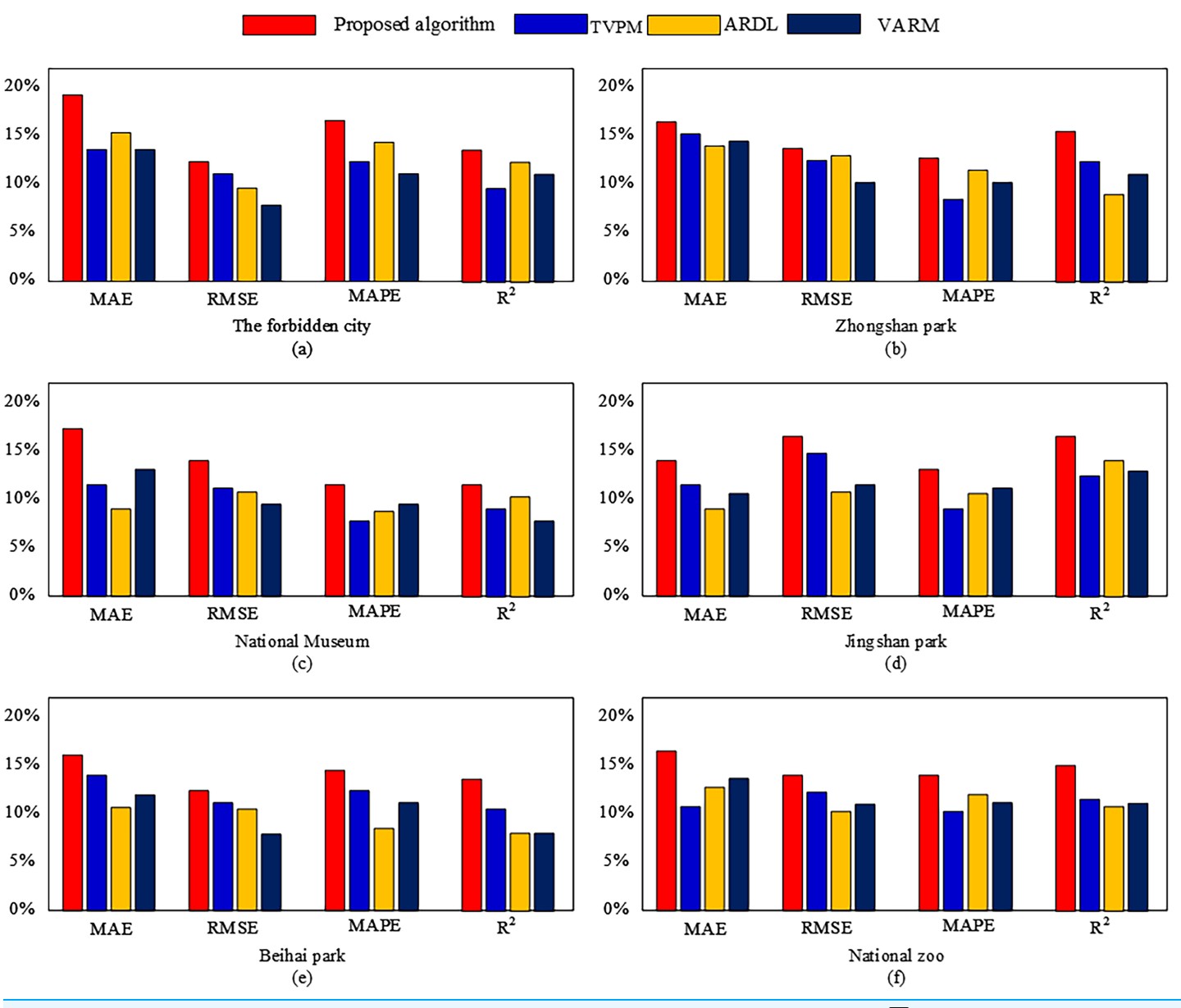

**Figure 7 (A–F) Predictive indicator results under different methods.**

Finally, we added different types of noise to the data to validate the robustness of the proposed algorithm, as shown in Tables 4 and 5.

By conducting stability analysis on different datasets and practical application scenarios, the performance and robustness of the algorithm are evaluated to ensure that it can maintain good performance and universality in various situations. This includes comparing the performance of models on different datasets, analyzing their stability in extreme situations, and evaluating their adaptability in different environments.

We conducted ablation experiments on the dataset to validate the effectiveness of our algorithm. This experiment modified the original model by removing only the self

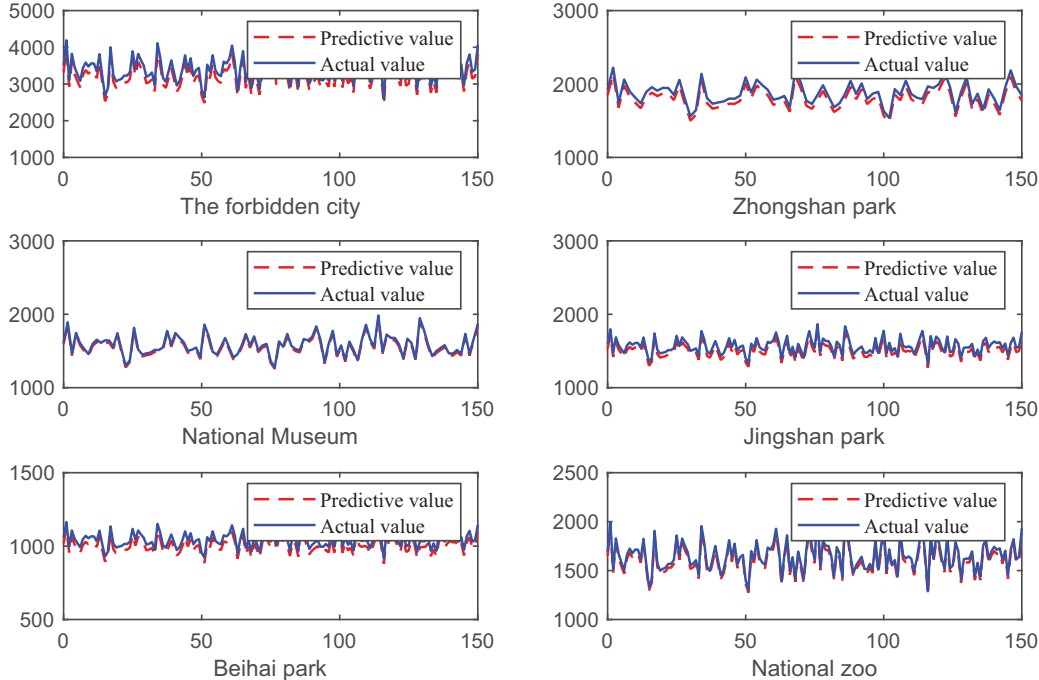

**Figure 8** The prediction performance of different scenic spots under the proposed algorithm.

**Table 2 Comparison of experimental results of various algorithms.**

| Dataset | The forbidden city | | Zhongshan park | | National museum | |
|---|---|---|---|---|---|---|
| | AUG | NDCG | AUG | NDCG | AUG | NDCG |
| Proposed algorithm | 0.5893 | 0.492 | 0.6014 | 0.511 | 0.6547 | 0.521 |
| PMF | 0.5914 | 0.509 | 0.6123 | 0.512 | 0.6612 | 0.539 |
| NCF | 0.6179 | 0.517 | 0.6219 | 0.519 | 0.6602 | 0.531 |
| SBPR | 0.6238 | 0.511 | 0.6418 | 0.507 | 0.6713 | 0.537 |
| AEASR | 0.6467 | 0.519 | 0.6556 | 0.527 | 0.6715 | 0.541 |

**Table 3 Comparison of experimental results of various algorithms.**

| Dataset | Jingshan park | | Beihai park | | National zoo | |
|---|---|---|---|---|---|---|
| | AUG | NDCG | AUG | NDCG | AUG | NDCG |
| Proposed algorithm | 0.5624 | 0.482 | 0.6029 | 0.501 | 0.669 | 0.526 |
| PMF | 0.5587 | 0.502 | 0.6119 | 0.514 | 0.682 | 0.532 |
| NCF | 0.6219 | 0.514 | 0.6217 | 0.529 | 0.692 | 0.521 |
| SBPR | 0.6118 | 0.512 | 0.6618 | 0.517 | 0.693 | 0.517 |
| AEASR | 0.6297 | 0.518 | 0.6754 | 0.518 | 0.686 | 0.521 |

**Table 4 Comparison of experimental results of various algorithms under increase disturbance 20%.**

| Dataset | The forbidden city | | Zhongshan park | | National museum | |
|---|---|---|---|---|---|---|
| | AUG | NDCG | AUG | NDCG | AUG | NDCG |
| Proposed algorithm | 0.5833 | 0.580 | 0.6923 | 0.561 | 0.6894 | 0.621 |
| PMF | 0.7914 | 0.789 | 0.7343 | 0.672 | 0.7542 | 0.739 |
| NCF | 0.8129 | 0.719 | 0.7239 | 0.669 | 0.7533 | 0.737 |
| SBPR | 0.7843 | 0. 711 | 0.7458 | 0.677 | 0.7713 | 0.739 |
| AEASR | 0.7808 | 0.739 | 0.7559 | 0.687 | 0.7715 | 0.761 |

**Table 5 Comparison of experimental results of various algorithms under increase disturbance 50%.**

| Dataset | The forbidden city | | Zhongshan park | | National museum | |
|---|---|---|---|---|---|---|
| | AUG | NDCG | AUG | NDCG | AUG | NDCG |
| Proposed algorithm | 0.6033 | 0.610 | 0.7526 | 0.601 | 0.7094 | 0.621 |
| PMF | 0.8924 | 0.839 | 0.8343 | 0.772 | 0.8592 | 0.819 |
| NCF | 0.8891 | 0.815 | 0.8289 | 0.661 | 0.8539 | 0.827 |
| SBPR | 0.8849 | 0. 811 | 0.8498 | 0.774 | 0.8763 | 0.831 |
| AEASR | 0.8828 | 0.839 | 0.8459 | 0.789 | 0.8815 | 0.803 |

**Table 6 Comparison of ablation experiment results.**

| Dataset | The forbidden city | | Zhongshan park | | National museum | |
|---|---|---|---|---|---|---|
| | AUG | NDCG | AUG | NDCG | AUG | NDCG |
| MSAMR-DSP | 0.5833 | 0.580 | 0.693 | 0.561 | 0.684 | 0.621 |
| MSAMR-DSP1 | 0.6478 | 0.621 | 0.709 | 0.601 | 0.702 | 0.692 |
| MSAMR-DSP2 | 0.6929 | 0.601 | 0.719 | 0.629 | 0.713 | 0.711 |

attention module, labeled as MSAMR-DSP1, and only the dynamic perception module, labeled as MSAMR-DSP2. The experimental results are shown in Table 6. Through ablation experiments, it was found that the method proposed in this article exhibited better performance.

## Experimental discussion

From the above experimental results, we can draw the following conclusion.

From Fig. 6, it can be seen that the algorithm proposed in this article has adaptive weight parameters. During the training process, the parameters are constantly changing, which can help the algorithm converge better.

From Fig. 7, it can be seen that the algorithm proposed in this article has significant advantages in MAE, RMSE, MAPE, and R metrics. This indicates that the proposed algorithm can better predict the number of tourists to tourist attractions, as shown in Fig. 8. In Fig. 8, we used the proposed algorithm to predict the future tourism volume of

different scenic spots, and the results showed that the predicted value can well predict the actual value.

Finally, from Tables 3 and 4, it can be seen that the algorithm proposed in this article has a high advantage in recommendation performance. The spatial optimization algorithm proposed in this article, which integrates multi-layer attention mechanisms, outperforms other baseline algorithms, especially traditional models such as PMF, SocialMF, and SBPR, in terms of various evaluation metrics on both datasets. This further proves that the fusion of attention mechanisms and recommendation systems has brought significant performance improvements.

## CONCLUSION

This study explores the development of smart tourism management by introducing a hierarchical attention mechanism to optimize spatial layout. The experimental results show that the algorithm performs well in tourism demand prediction, tourist behavior analysis, and scenic spot recommendation. The introduction of hierarchical attention mechanism has improved the performance and efficiency of the system, providing new methods and ideas for smart tourism management. Comprehensive analysis shows that combining attention mechanisms with spatial layout optimization plays an important role in promoting smart tourism management, bringing new opportunities and challenges to the development of the tourism industry.

## ACKNOWLEDGEMENTS

We thank the anonymous reviewers whose comments and suggestions helped to improve the manuscript.

### Funding
This work was supported by the MOE (Ministry of Education in China) Project of Humanities and Social Sciences (22YJA760106), the Key Project on Higher Education Reform Research in Jiangsu Province (2023JSJG216), and the Project of Jiangsu Tourism and Culture Research Institute (24jslwy007). The funders had no role in study design, data collection and analysis, decision to publish, or preparation of the manuscript.

### Grant Disclosures
The following grant information was disclosed by the authors:
MOE (Ministry of Education in China) Project of Humanities and Social Sciences: 22YJA760106.
Key Project on Higher Education Reform Research in Jiangsu Province: 2023JSJG216.
Project of Jiangsu Tourism and Culture Research Institute: 24jslwy007.

### Competing Interests
The authors declare that they have no competing interests.

## Author Contributions

- Jie Ding conceived and designed the experiments, performed the computation work, prepared figures and/or tables, authored or reviewed drafts of the article, and approved the final draft.
- Lingyan Weng performed the experiments, analyzed the data, performed the computation work, prepared figures and/or tables, authored or reviewed drafts of the article, and approved the final draft.
- Lili Fan analyzed the data, performed the computation work, prepared figures and/or tables, authored or reviewed drafts of the article, and approved the final draft.
- Peixue Liu performed the experiments, performed the computation work, prepared figures and/or tables, authored or reviewed drafts of the article, and approved the final draft.

## Data Availability

The data and code are in the Supplemental File.

The data is also available at Zenodo: Yang, X., Wang, L., Ma.Pengfei, He, Y., Zhao, C., & Zhao, W. (2023). Urban and suburban decadal variations in air pollution of Beijing and its meteorological drivers [Data set]. Zenodo. https://doi.org/10.5281/zenodo.8188665.

## Supplemental Information

Supplemental information for this article can be found online at http://dx.doi.org/10.7717/peerj-cs.2329#supplemental-information.

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
