# Peer review of "Spatial layout optimization model integrating layered attention mechanism in the development of smart tourism management"

_PeerJ Computer Science, doi:10.7717/peerj-cs.2329_

## Round 0.1 · original submission · Major Revisions

Please see both R1 and R2 detailed reviews. Key recommendations include ensuring objective language and detailed implementation descriptions, especially for new mechanisms like multi-layer self-attention and dynamic perception algorithms. Overall, the guidelines emphasize clarity, detailed methodology, and thorough explanation of results to ensure the validity and utility of the findings.

Reviewer 1 ·

Basic reporting

The authors use the user-based self-attention network module to dynamically learn the user's interest in the attraction at different time points, and use the attraction-based self-attention network module to search for potential feature transformation patterns between attractions. My Suggestions for improvements include describing implementation details in detail, extending the experimental design, performing algorithm interpretation and stability analysis along with the following:

Please ensure that the language of the manuscript is objective, For example, "To the best of the author's knowledge, not many research have used geographical data to estimate traveler demand. Spatial lag factors have been included into time series or econometric models in five studies to create spatiotemporal autoregressive models. "What are five studies? I don't see the corresponding reference;

What are dynamic perception algorithms? The author adopts many general concepts without clearly defining and explaining them.

Experimental design

Ensure that for each new mechanism introduced in the paper, especially the multi-layer self-attention mechanism and the dynamic perception algorithm, there is enough focus, that is, explain in detail how to capture the complex relationship between users and travel destinations through these mechanisms, and explain how these relationships affect the final tourism prediction results;

Multi-layer self-attention mechanisms and dynamic perception algorithms are mentioned in the paper, but more detail is needed on how these mechanisms are implemented in the model. Specifically, you need to clearly list the actions and parameter Settings for each layer;

How the model adjusts the weights according to different temporal and spatial variations of users and travel destinations. At the same time, stability analysis, that is, how the algorithm performs in different data sets and situations, is also needed to ensure its robustness and versatility in practical applications.

Equation (12) normalizes the alignment scores using the softmax function. Ensure the denominator's notation is clear and accurately represents the sum over all trusted users s_u for user u.

Validity of the findings

Include a clear visualization or schematic (like Figure 4) to illustrate the network structure and flow of information, enhancing understanding of how inputs are processed through the MLP layers to generate the final recommendations.

English expressions need to be further optimized, e.g “where Among them, o is the output of the network before residual connection, and e is the embedding.

Reviewer 2 ·

Basic reporting

How the model adjusts the weights according to different temporal and spatial variations of users and travel destinations. At the same time, stability analysis, that is, how the algorithm performs in different data sets and situations, is also needed to ensure its robustness and versatility in practical applications.

The literature review section lists a large number of econometric model-oriented forecasting models, The relevant research literature is cited to explain the theoretical basis and previous research results of these technologies in tourism forecasting or related fields.

Explain why these methods can improve the accuracy and utility of tourism prediction.

A flow chart/pseudocode is added to clearly explain the Multilayer self-attention mechanism recommendation algorithm based on dynamic spatial perception MSAMR-DSP;

Experimental design

The choice of activation functions (ReLU) in MLPs is discussed due to concerns over gradient vanishing and computational efficiency. While ReLU helps mitigate gradient issues, ensure the reasons for choosing it over alternatives (like sigmoid or tanh) are well-justified based on the model's specific requirements, such as handling non-linearities and network depth.

Alignment Score Calculation: Equation (11) defines the alignment score between a user u and its trusted user m. To enhance clarity, explicitly state the dimensions of W_1, W_2, b_1, and b_2. Additionally, clarify how ReLU is applied within the MLP layers and how it contributes to the alignment score computation.

Comprehensive User Representation: Equation (13) defines P ˜_u, the comprehensive feature representation of user u, combining its original feature p_u with weighted sums of its trusted users' embeddings s_m. Clarify the parameter α and its role in balanced between the user's original features and the trusted user embeddings.

Validity of the findings

Alignment Score Calculation: Equation (11) defines the alignment score between a user u and its trusted user m. To enhance clarity, explicitly state the dimensions of W_1, W_2, b_1, and b_2. Additionally, clarify how ReLU is applied within the MLP layers and how it contributes to the alignment score computation.

Additional comments

Given in basic reporting

---

## Round 0.2 · accepted · Accept

I confirm that the authors have addressed all of the reviewers' comments.

Reviewer 1 ·

Basic reporting

no comment

Experimental design

no comment

Validity of the findings

no comment

Reviewer 2 ·

Basic reporting

The authors have incorporated all the provided suggestions.

Experimental design

The authors have incorporated all the provided suggestions.

Validity of the findings

The authors have incorporated all the provided suggestions.

Additional comments

No